# A Prospective Study of Training Methods for Two-Year-Old Thoroughbred Racehorses in Queensland, Australia, and Analysis of the Differences in Training Methods between Trainers of Varying Stable Sizes

**DOI:** 10.3390/ani11040928

**Published:** 2021-03-25

**Authors:** Kylie L. Crawford, Anna Finnane, Ristan M. Greer, Clive J. C. Phillips, Emma L. Bishop, Solomon M. Woldeyohannes, Nigel R. Perkins, Benjamin J. Ahern

**Affiliations:** 1School of Veterinary Science, The University of Queensland, Gatton 4343, Australia; s.woldeyohannes@uq.edu.au (S.M.W.); n.perkins1@uq.edu.au (N.R.P.); b.ahern@uq.edu.au (B.J.A.); 2School of Public Health, The University of Queensland, Herston 4006, Australia; a.finnane@uq.edu.au; 3Torus Research, Bridgeman Downs 4035, Australia; rmg@torusresearch.com.au; 4School of Medicine, The University of Queensland, Herston 4006, Australia; 5Curtin University Sustainability Policy (CUSP) Institute, Curtin University, Perth 6845, Australia; Clive.Phillips@curtin.edu.au; 6Garrards Equine Veterinary Practice, Albion 4010, Australia; bishope@garrards.com.au

**Keywords:** racehorse, thoroughbred, training, rest period, two-year-old

## Abstract

**Simple Summary:**

Musculoskeletal injuries present significant concerns for the global Thoroughbred racing industry. The development of training and management strategies to reduce injuries has been challenging due to conflicting findings about the risk factors for injury, and limited understanding of the role of different training methodologies. There is little published data on patterns of rest periods and exercise data and how these vary between trainers and between different racing jurisdictions. We describe training methodologies for 535 two-year-old horses providing 1258 training preparations and 7512 weeks of exercise. We investigated the variation in training methodologies between trainers from small, medium and large stable sizes. Significant differences were observed between trainers, with horses from larger stables accumulating a greater high-speed exercise volume, attaining training milestones more frequently and taking less time to reach their training milestones. We also highlight similarities and differences in training methods used in Queensland, Australia, and those previously reported from other geographic locations. A better understanding of training methods is an essential step towards reducing the impact of injuries.

**Abstract:**

Worldwide, musculoskeletal injuries remain a major problem for the Thoroughbred racing industry. There is a strong interest in developing training and management strategies to reduce the impact of musculoskeletal injuries, however, progress has been limited by studies reporting conflicting findings, and a limited understanding of the role of different training methods in preventing injury. There is little data on patterns of rest periods and exercise data and how these vary between trainers. This prospective study of two-year-old racehorses was conducted in Queensland, Australia and involved weekly personal structured interviews with 26 trainers over 56 weeks. Detailed daily exercise data for 535 horses providing 1258 training preparations and 7512 weeks at risk were collected. Trainers were categorised into three groups by the mean number of two-year-old horses that they had in work each week over the study duration: (1) Small stables with five or less, (2) Medium stables with 6 to 15 and (3) Large stables with greater than 15 horses in training. Differences between trainers with small, medium and large stable sizes were evaluated using linear regression, Kruskal–Wallis equality-of-populations rank test if linear models were mis-specified or Chi-squared tests for categorical variables. Significant differences were observed between trainers, with horses from larger stables accumulating a greater high-speed exercise volume (*p* < 0.001), attaining training milestones more frequently (*p* = 0.01) and taking less time to reach their training milestones (*p* = 0.001). This study provides detailed data to which training practices from other locations can be compared. Presenting actual training data rather than trainers’ estimation of a typical program provides a more accurate assessment of training practices. Understanding how training practices vary between regions improves comparability of studies investigating risk factors and is an important step towards reducing the impact of musculoskeletal injuries.

## 1. Introduction

Worldwide, musculoskeletal injuries (MSI) remain a significant problem for the Thoroughbred racing industry [1,2,3,4,5,6,7,8,9,10]. Common types of MSI reported in Thoroughbred racehorses include lower limb fractures [11,12,13,14,15,16,17,18,19,20], tendon and ligament injuries [11,12,14,15,16,17,21,22,23,24,25], carpal and fetlock osteochondral fragments [11,14,15,16,17,26,27] and dorsal metacarpal disease [14,15,26,28,29,30,31]. Dorsal metacarpal disease is the most common MSI in two-year old horses in Queensland, Australia, followed by carpal osteochondral fragments and traumatic injuries or lacerations [32]. Musculoskeletal injuries are predominantly repetitive stress injuries of bones, tendons, and ligaments whereby cumulative loading results in tissue fatigue and ultimately failure [25,33,34,35,36]. Repetitive loading of bone results in bone fatigue and microdamage in the form of microscopic linear cracks [36,37,38,39]. Once bone microdamage has occurred, continued loading beyond the strength of the bone results in coalescing of microcracks into stress fracture lines [36,40,41,42]. Loading of microcracks can also result in complete bone failure (catastrophic fractures) [36,40,41,42]. Repetitive loading of tendons and ligaments results in tissue fatigue in the form of degenerative changes in the collagen fibrils and extracellular matrix in the central region [35,43,44]. These degenerative changes appear histologically as decreased fibril diameter and a reduced crimp angle to the tendon structure [25,43,44].

There has been, and continues to be, a strong interest in developing training and management strategies to reduce the impact of MSI. Training-related risk factors associated with MSI include the volume, intensity and rate of accumulation of high-speed exercise [12,15,18,19,31,45,46,47,48], the volume of low-speed exercise [18,31,49], the interaction between low and high-speed exercise [18,31], the training preparation length [15,50,51] and the length of rest periods between training preparations [18,31,49,52,53,54,55]. The risk of MSI also varies between trainers [19,30,51,56,57,58].

Developing training and management strategies to mitigate the risk of MSI has been limited by conflicting findings for risk factors of MSI and a limited understanding of training methodologies. There is little published data on patterns of rest periods and exercise data and how these vary between trainers and between different racing jurisdictions. Numerous studies have examined various aspects of training methodologies. However, studies typically reported different outcome measures or had low numbers, did not provide detailed information on both exercise and rest periods or examine the differences between trainers [59,60,61,62]. A survey of 66 trainers in Victoria, Australia recently described typical training methods in detail, but this does not take into account the involuntary interruptions due to medical issues, performance, appetite and track conditions [63]. Bolwell and colleagues have reported detailed training methods of New Zealand horses, the variation between trainers and the effect of voluntary and involuntary interruptions, yet it is unclear whether those findings are applicable to other geographic locations [14,64,65,66]. There is a need for further data on what constitutes an actual training program, once voluntary and involuntary interruptions have been accounted for, rather than trainers’ reporting of a typical or intended training program. Further information on rest periods, training intensity and cumulative high-speed weekly exercise volumes and attaining milestones is required.

This study addressed these research gaps through a prospective cohort study in two-year-old racehorses training in Queensland, Australia. We were particularly interested in the training volume and preparation characteristics of two-year-old horses because a higher incidence of MSI has been reported in two-year-old horses [26,30,49,67,68] and there is evidence that the training program can be modified to improve skeletal adaptation and potential resilience to MSI [62,69,70,71,72,73,74,75,76,77]. Our aims were to: (1) Determine the actual two-year-old training volume and preparation characteristics for these horses and (2) Examine whether the training volume and preparation characteristics vary between trainers of differing stable sizes.

## 2. Materials and Methods

### 2.1. Recruitment of Participants

This study was performed concurrently with a study investigating the incidence and type of MSI in Thoroughbred racehorses in South-East Queensland, Australia, which describes our recruiting process in detail [32]. Although recruited trainers had a minimum of three horses in training at the time of recruitment, we did not specify a minimum number of two-year-old horses in training. Human (2017001248) and animal (SVS/384/17) ethics approval was obtained from the University of Queensland Ethics Committee. Trainers from three metropolitan racecourses that agreed to participate in the study enrolled all two-year-old horses under their care.

### 2.2. Study Design

A prospective survey, whereby weekly personal structured interviews (Appendix A
Figure A1) were conducted with participating trainers or their forepersons to collect detailed training and exercise data over a 13-month (56 week) period. Structured personal interviews facilitated expedient collection of complete, detailed data and enabled clarification of any inconsistencies encountered.

### 2.3. Data Collection

#### 2.3.1. Classification of Trainers

Trainers were categorized into three groups according to the average number of two-year-old horses that they had in training: (1) Small stables with five or less, (2) Medium stables with 6 to 15 and (3) Large stables with greater than 15 horses in training.

Traditionally, registered trainers in Queensland are classified as (1) Class A (most experienced), (2) General or (3) Restricted (least experienced or have restrictions imposed for reasons including holding a concurrent jockey license). Classification of trainers according to their apparent expertise was deemed inappropriate for this study. The Queensland Racing Trainer premiership list revealed 14/20 of the leading trainers in Queensland [78] were not Class A trainers [79]. Thus, we felt that holding a Class A license did not necessarily reflect the most expert or successful trainers, and classification according to stable size was a more useful method.

#### 2.3.2. Two-Year-Old Horses

A horse was considered to be a “two-year-old” until 1 August of its third year of life. August first is the date where Thoroughbred horses in Australia officially increase a year of age regardless of their actual date of birth. This definition incorporates all racehorses two years of age and younger, as racehorses in Australia are usually younger than two years when they commence race training. Horse identity and age were confirmed with the Australian Stud Book [80]. All two-year-old horses under the care of the licensed trainer were enrolled. Trainers were not able to select which horses contributed data.

#### 2.3.3. General Racing and Training History

The age that the horse started training (if known), the number of training preparations (period of exercise uninterrupted by rest) that the horse had previously completed, and the time spent in pre-training prior to the current preparation was collected from the trainer or foreperson when the horse entered the study. The rest period before the preparation was not obtained for horses new to the study, as this was often unknown, and was deemed inaccurate. For horses re-entering the study after a rest, the rest period was calculated from the time of exit.

#### 2.3.4. Detailed Daily Exercise Data

Detailed daily exercise data were collected for all horses every week for the 56-week study period. Horses were censored on August 1, when they turned three years of age, if they left the trainer and at study completion.

The number of days of low-speed exercise were recorded.

Slow days were defined as when the horse was exercised at speeds of less than 15 s/furlong (13 m/s; 800 m/min; 48 km/h). The number of days of non-ridden exercise were also recorded.

Walking was defined as when horses were only exercised on the walking machine or walked by hand. This did not incorporate those horses undertaking warm-up exercise on the walker prior to exercise on the racetrack, nor those that were exercised on the walker in the afternoon in addition to their morning exercise at the racetrack.Water-walkers are defined as walking machines in a shallow swimming pool, with the water up to approximately the level of the horses’ chest.Swimming was defined as exercise in a swimming poolTreadmills are defined as stationary exercise machines with continuous belts that facilitate exercise at low or high speeds with or without an incline.The number of days and the distances (furlongs) of high-speed exercise were also recorded.

The number of days of high-speed exercise were also recorded High-speed exercise variables included:Three-quarter pace: defined as 15 s/furlong (13 m/s; 800 m/min; 48 km/h)Track gallop: gallop exercise performed during track-work, defined as faster than 13 s/furlong (15 m/s; 900 m/min; 55 km/h)Jump-outs (non-official trials)Official trialsRaces (official trial and race data were cross-checked with the RA public database [81]).

Total exercise at a gallop was calculated by adding track gallop, jump-out, trials and races.

#### 2.3.5. Total and Average Cumulative Days and Distances

The cumulative days and distances in furlongs (200 m) were calculated by adding the days and distances, respectively, from the detailed daily exercise data. The average days and distances were calculated by dividing the total days and distances, respectively, by the total number of weeks in training.

#### 2.3.6. Categories of Preparations and Trials

A training preparation was defined as the period of time that the horse was undertaking race training in the racing stables. This did not incorporate any pre-training that was conducted elsewhere. A preparation was considered to have ended when training did not occur for seven days. Rest periods occurred between training preparations. The training preparations that horses undertook were numbered sequentially from the time that the horse began race training. We categorised the number of preparations into three groups (1) The first preparation, (2) The second preparation and (3) the third and subsequent preparations. The training volumes were deemed equivalent for the third and subsequent preparations.

The training volumes were also categorised into (1) up to the first official trial and (2) after the first official trial. An official race trial involves horses ridden by professional jockeys, jumping out of the barriers and racing under timed conditions. These trail races are broadcast and results are publicly available. When horses had not completed an official trial, the time of first race was used instead. At the time of the study, it was not compulsory for Thoroughbred racehorses in Queensland to undertake an official trial prior to racing. A barrier certificate could be obtained from their performance in jump-outs (unofficial trials). Jump-outs involve horses ridden by either professional jockeys or licenced trackwork riders and are unofficially timed and results are not publicly available. Scheduling of trials and track conditions frequently dictated whether trainers chose not to participate in official trials. Additionally, some trainers chose not to participate in official trials as these trials are broadcast nation-wide and this could influence the totalisator agency board odds and potential gambling revenue. Jump-outs were not broadcast and there is no legislation requiring the jockey or licensed trackwork rider to ride the horse to its full potential, thus the trainer could obtain the barrier certificate for the horse and gain an insight into its potential ability, without this being disclosed to the gambling public.

### 2.4. Data Analysis

All analyses were performed using Stata 15.1^®^ (Statacorp, College Station, TX, USA). Normality of continuous data were assessed using histograms. Normally distributed data were presented as mean and standard deviation. Non-normally distributed data were presented as median and interquartile range. If the 25% and 75% quartiles were both zero, the range (minimum and maximum values) were presented in parentheses instead. Categorical data were presented as numbers and percentages.

For continuous variables, differences between trainers with small medium and large stables were assessed using linear regression. Model specification was checked by examining histograms of the raw residuals and plots of the predicted probabilities against the standardized residuals [82]. A Kruskal–Wallis equality-of-populations rank test was used if linear models were mis-specified. For categorical variables, differences between trainers with small, medium and large stables were assessed using Chi-squared tests. Significance was set at α = 0.05 for all tests.

## 3. Results

The trainers who participated in this study were the same population who contributed to a concurrent study of musculoskeletal injuries and trainer characteristics have been described in detail previously [32]. Briefly, 27 out of 40 eligible trainers (68%) agreed to participate. Data were collected every week for 56 weeks (13 months) from November 2017 to December 2018 for 26/27 (96%) of trainers. One trainer did not train any two-year-old horses for the study duration. Another trainer only contributed six months of data before retiring from training. Trainers provided exercise data for 535 two-year-old horses, who completed 1258 training preparations over 7512 weeks. Individual trainers had between 1 and 43 (mean = 15, SD = 10) two-year-old horses in training.

Training and racing data for the 535 horses summarised by trainer are presented in Appendix A
Table A1. Data stratified by preparation number (whereby 377, 350 and 349 horses contributed data for preparations 1, 2 and 3 onwards, respectively) are presented in Table 1, Table 2 and Table 3.

### 3.1. General Training Characteristics 

The general training characteristics are presented in Table 1, Table 2 and Table 3. Trainers with large stables (>15 two-year-old horses in training) and medium stables (6–15 two-year-old horses) started training their horses earlier than small stables (5 or less two-year-old horses) (*p* < 0.001). Horses from large stables also had a higher number of training preparations (*p* < 0.001). Horses from large stables had shorter preparation lengths for preparation numbers 1 (*p* < 0.001). and 2 (*p* < 0.001). There were no significant differences between trainer categories for preparations 3 onwards. Trainers with large stables gave their horses less pre-training overall than trainers with small and medium stables (*p* < 0.001).

### 3.2. Low-Speed Exercise 

The low-speed exercise characteristics are presented in Table 1, Table 2 and Table 3. Overall, the total cumulative days exercised at a slow pace decreased with stable size (*p* < 0.001). An equivalent pattern was seen when analyses were repeated for average days per week and for individual training preparations.

### 3.3. Non-Ridden Exercise 

The non-ridden exercise characteristics are presented in Table 1, Table 2 and Table 3. Trainers with large stables also reported a higher cumulative number of days exercised on walkers (*p* < 0.001), water-walkers (*p* < 0.001), and treadmills (*p* < 0.001), whereas trainers with medium stables reported a higher cumulative number of days swimming (*p* < 0.001). Equivalent patterns were seen when analyses were repeated for average number of days per week in training and by preparations.

### 3.4. High-Speed Exercise 

The high-speed exercise characteristics are presented in Table 1, Table 2 and Table 3.

#### 3.4.1. Total Cumulative Days

Overall, horses trained in large stables had a higher number of cumulative days exercised at three-quarter pace (*p* = 0.01), track gallops (*p* < 0.001), trials (*p* < 0.001), and total days gallop (*p* < 0.001). An equivalent pattern was observed for the overall average of days per week in training, but not when analyses were repeated by training preparation.

For preparation number 1, the number of cumulative days of jump-outs (*p* = 0.03) and trials (*p* = 0.01) was highest for small stables.

For preparation number 2, the number of cumulative days exercised at three-quarter pace was highest for small stables (*p* = 0.003) and the number of jump-outs (*p* < 0.001). The number of days racing were highest for medium stables (*p* = 0.002).

For preparations 3 onwards, the number of cumulative days exercised at three-quarter pace (*p* = 0.003), track gallop (*p* = 0.01), jump-outs (*p* = 0.002), races (*p* < 0.001) and total days at a gallop (*p* = 0.003) were highest for medium stables. For each preparation category, similar patterns were observed for average days per week in training.

#### 3.4.2. Total Cumulative Distance (furlongs, 200 m)

Overall, horses trained out of large stables had the highest cumulative number of furlongs exercised during track gallops (*p* < 0.001), trials (*p* < 0.001), and total furlongs at a gallop (track gallops, jump-outs, trials and races) (*p* < 0.001). A similar pattern was observed for average distance per week in training.

For preparation number 1, the number of cumulative furlongs of trials was highest for horses from small stables (*p* = 0.02). A similar pattern was observed for the average distance per week.

For preparation 2, horses trained from small stables had a higher number of furlongs exercised at three-quarter pace (*p* < 0.001) and horses trained by medium trainers had higher distances accumulated in jump-outs (*p* < 0.001), races (*p* = 0.002) and total distances galloped (*p* = 0.04). Equivalent patterns were observed for the average distances per week in training.

For preparations 3 and onwards, horses trained from medium stables had the highest cumulative distances of three-quarter pace (*p* < 0.001), track gallop (*p* < 0.001), jump-outs (*p* = 0.002), races (*p* < 0.001) and total distance galloped (*p* = 0.01). An equivalent pattern was observed for the average distances reported per week.

### 3.5. Reached Training and Racing Milestones 

The numbers of horses reaching the training and racing milestones are presented in Table 1, Table 2 and Table 3. Larger stables had the highest percentage of horses that ever galloped (*p* < 0.001) and trialled (*p* < 0.001) during the study.

During preparation 1, medium stables had the highest percentage of horses that trialled (*p* = 0.02). During preparations 2 and 3, medium stables had the highest percentage of horses jump-out (*p* < 0.001, *p* = 0.02) and race (*p* = 0.01, *p* = 0.001).

### 3.6. Time to Reach Training and Racing Milestones

The times for horses to reach the training and racing milestones are presented in Table 1, Table 2 and Table 3. Overall, horses trained out of larger stables took significantly less time to reach three-quarter pace (*p* < 0.001), gallop (*p* < 0.001), jump-out (*p* = 0.01), trial (*p* < 0.001) and race (*p* = 0.01).

During preparation number 1, horses trained from larger stables took significantly less time to reach three-quarter pace (*p* = 0.001). During preparation number 2, horses trained from larger stables took significantly less time to reach three-quarter pace (*p* < 0.001), gallop (*p* < 0.001), jump-out (*p* < 0.001) and trial (*p* < 0.001). There were no significant differences between trainer categories in the time taken to reach milestones in preparations 3 onwards.

### 3.7. Median Cumulative Weekly Training Volume to First Trial

The median cumulative weekly high-speed exercise training volumes for two-year-old horses up until the time of their first official race trial are summarized below in Figure 1.

### 3.8. Median Cumulative Weekly Training Volume after First Trial

The median cumulative weekly high-speed exercise training volumes for two-year-old horses after the time of their first official race trial are summarized below in Figure 2.

## 4. Discussion

This paper presents detailed data on the characteristics of training preparations in a population of two-year-old horses in Queensland, Australia. The median preparation length for this group of horses (overall preparations 7 weeks, IQR 4-11) was similar to the two-year-old horses in New South Wales, Australia [60] but lower than that reported in New Zealand [62,83]. Trainers in the current study, especially the larger sized stables, typically introduced their horses to racing over two shorter preparations (preparation one 4 weeks, IQR 2-6, preparation two 6 weeks, IQR 4-10), rather than a single 13–15 week preparation as described by Rogers and Firth in New Zealand [62]. This may reflect regional differences in training methodologies.

In the current study, non-ridden exercise modalities were used increasingly as stable size increased. Other studies have reported similar use of non-ridden training modalities in racehorses training in New Zealand [64] and in Victoria, Australia [84]. Incorporating non-ridden exercise modalities into race training programs may be beneficial in reducing the risk of musculoskeletal injuries [64,65].

There were significant differences between trainers for the overall volume of high-speed exercise parameters in the current study. The overall total cumulative number of days and the distance exercised at three-quarter pace (15 s/furlong), track gallops (faster than 13 s/furlong), official barrier trials and total days galloping (combined track gallops, unofficial trials, official trials, and races) were significantly higher for horses from large stables.

The high-speed training volume in this study is similar to the “high-volume workload over extended time” type of two-year-old training program reported in Victoria, Australia [63]. Interestingly, the high-speed training volume for the first preparation was much smaller than that described in New Zealand [62]. This most likely reflects the increased number of shorter preparations used to introduce horses to racing in the current study. The high-speed training volume for the second training preparation is comparable to the first preparation [62] or up until the first trial [64,85] described in New Zealand. It is also similar to values reported for the first preparation to include exposure to high-speed exercise described in racehorses in New South Wales, Australia [60].

There were also significant differences in the proportion of horses reaching training milestones between trainers. Trainers running larger stables were more likely to have their horses reaching milestones. Overall, approximately 40% of the two-year-old horses in this study started in a trial and 30% started in a race. This is lower than previous studies which reported approximately 80% of two-year-old horses started in a trial [65] and approximately 50% started in a race [59,65,86]. The difference in findings likely reflect the populations of horses in the studies. All horses in one study and a high proportion of horses in the second study were from elite yearling sales, which were renowned for producing high-quality early two-year-old racehorses. Sale horses are more likely to reach the training milestones as two-year-old horses [65]. Details of the proportion of horses in the current study that were from these elite sales were not available. Alternatively, it is possible that many of the horses in the current population were not considered by trainers to be able to withstand the stresses of racing as two-year-old horses and were instead being trained towards racing as three-year-old horses. It is unlikely that a low proportion of horses in this population did not trial or race due to injury, as the incidence of injuries in this group of horses was low compared to other regions [32]. The low injury rate does support trainers in the current study taking a more conservative approach with horses not considered ready to trial or race. The time taken to reach the training milestones was also shorter for the trainers with larger stable sizes. Interestingly, the median times to the first start in an official trials and races were much shorter than those reported in prospective studies from New Zealand [61,65], but equivalent to responses from a survey of trainers in New Zealand [66] and Victoria, Australia [63]. The disparity in findings between the prospective and survey studies from New Zealand emphasizes the difference between the intended (“typical”) training programs reported by trainers, and the actual training programs that occur after voluntary and involuntary interruptions have occurred [14,65,66].

This study provides detailed data on training methodologies in this population of horses. The knowledge of what constitutes an actual training program, incorporating both planned and unplanned interruptions could be an important step towards understanding how a two-year-old training program can be modified to reduce the risk of musculoskeletal injury. By understanding what constitutes a realistic training volume at different stages of the training preparations and career, data from injury studies can be better applied to a training program.

This study also highlights the similarities and differences to the training methodologies reported in previous studies from other geographic locations. An awareness of the training practices for different geographic regions facilitates better utilization of data from other racing jurisdictions.

A strength of this study is detailed data collected by weekly personal interview, which reduces recall bias or missing data that results from trainers completing forms. Selection bias was also minimized by enrolling all of the horses under the trainers’ care, rather than allowing trainers to choose which horses contributed data to the study. It is also an accurate representation of the actual training program, rather than a survey of a “typical” training program, as these can vary substantially from the actual training program due to voluntary and involuntary interruptions [65,66,85].

The primary limitations of this study include the population of horses investigated, reporting of data and analysis using univariable rather than multivariable models. The population investigates reflects racehorses in South-East Queensland, Australia, and the results may not be generalizable to other geographic locations and populations of horses. Exercise data collected were reported by trainers and speeds were not validated using GPS or other modalities and values are therefore must be considered an approximation. Univariable analytical techniques were used because training practices including the amount of rest, intensity of track exercise, use of non-ridden exercise modalities, and starting a horse in a trial or race are determined by so many horse-related, environmental and other variables that we were unable to measure. Horse-related unmeasured variables include appetite, bodyweight, behaviour, health, injury, ability and fitness. Environmental and other unmeasured variables include temperature, rainfall, track condition and scheduling of suitable trials and races. Consequently, adjusting for only the available variables is not only inadequate, but it could also produce misleading associations. Thus, we sought to report the actual training practices rather than evaluate the drivers for these practices.

There are also limitations to data for horses obtaining the milestones of starting in an official trial and race because at the time of the study it was not compulsory for horses to compete in an official trial before they raced. Some trainers elected not to start their horse in official trials as the horses’ performance could affect the potential gambling prize money as well as international sale price. Consequently, the proportion of horses reaching their training milestones may be under-represented. Furthermore, the time to reach the milestones may be under-represented due to our definition of a preparation ending after no training exercise for 7 days. When horses were rested for short breaks, the time to reaching the milestones in the following preparation was markedly reduced. This is likely to be because no fitness was lost during the short rest period.

A further limitation is that some trainers used non-official trials (jump-outs) as barrier practice, and horses did not always reach full galloping speed. This would overestimate our cumulative total days and distance of gallop exercise. Finally, due to the timing of our study, we were unable to follow horses for their entire two-year-old season. Data were collected from November 2017 to December 2018. Consequently, we are only able to provide data for the 56 weeks of the study, not the entire two-year-old season. The two-year-old season begins on 1 August, thus the horses born in 2015 did not contribute data from August 2017 until data collection began in November 2017 and horses born in 2016 did not contribute data after collection finished in December 2018. Numbers of two-year-old horses were approximately equal for the 2017 and 2018 seasons.

Finally, the current study did not evaluate the use of adjunctive methods reported for monitoring the training of athletic horses, including blood lactate concentrations [87,88], serum biomarkers or cytokines [89,90,91,92] and peripheral blood mononuclear cells [93,94] as this data were not available. Future research evaluating how training methodology affects these monitoring tools would be useful.

## 5. Conclusions

This paper provides detailed data on training methodologies for this jurisdiction in a manner that is easily interpreted by industry stakeholders. We highlight similarities and differences in methodologies previously reported from other geographic locations. An understanding of actual training methodologies is an important step towards reducing the risk of musculoskeletal injuries.

## Figures and Tables

**Figure 1 animals-11-00928-f001:**
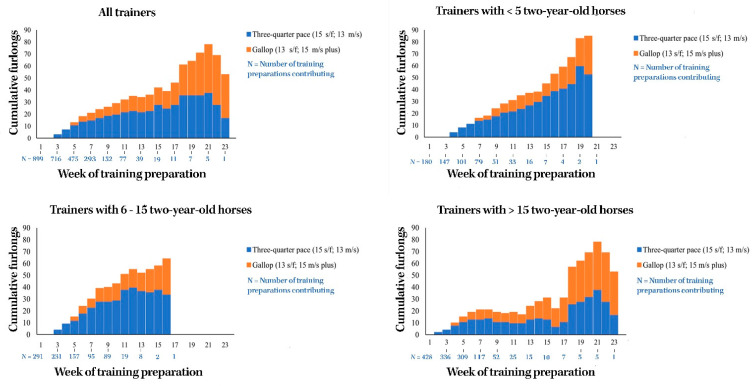
The median cumulative weekly high-speed exercise training volumes for two-year-old Thoroughbred racehorses up until the time of their first official race trial in Queensland, Australia.

**Figure 2 animals-11-00928-f002:**
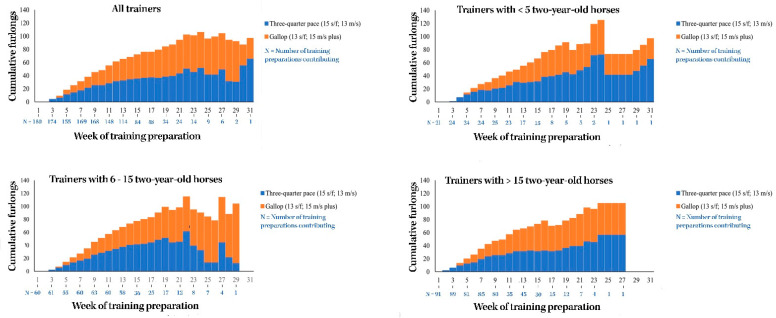
The median cumulative weekly high-speed exercise training volumes for two-year-old Thoroughbred racehorses after their first official race trial in Queensland, Australia.

**Table 1 animals-11-00928-t001:** Summary training and racing statistics for two-year-old horses in their first training preparation in Queensland, Australia.

		All Two-Year-Old Horses		Small Stables		Medium Stables		Large Stables		*p*-Value ‡
		(n = 377)		(n = 96)		(n = 152)		(n = 129)		
General training characteristics										
		Median	IQR	Median	IQR	Median	IQR	Median	IQR	
Rest period before (wks)		†	†	†	†	†	†	†	†	†
Pre-training (wks)		†	†	†	†	†	†	†	†	†
Prep length (wks)		4	2–6	4	3–10	4	2–7	3	2–4	<0.001
Low speed exercise										
Total cumulative days										
		Median	IQR ^¥^	Median	IQR ^¥^	Median	IQR ^¥^	Median	IQR ^¥^	
Slow		17	10–28	24	16–45	19	12–28	10	4–16	<0.001
Average days per week in training										
		Median	IQR ^¥^	Median	IQR ^¥^	Median	IQR ^¥^	Median	IQR ^¥^	
Slow		5	4–5	5	5–6	5	4–6	4	3–4	<0.001
Non-ridden exercise										
Total cumulative days										
		Median	IQR ^¥^	Median	IQR ^¥^	Median	IQR ^¥^	Median	IQR ^¥^	
Non-ridden		0	0–1	0	(0–24)	0	(0–72)	0	0–1	0.02
Average days per week in training										
		Median	IQR ^¥^	Median	IQR ^¥^	Median	IQR ^¥^	Median	IQR ^¥^	
Non-ridden		0	0–0.05	0	(0–6)	0	(0–4)	0	0–0.3	0.01
High-speed exercise										
Total cumulative days										
		Median	IQR ^¥^	Median	IQR ^¥^	Median	IQR ^¥^	Median	IQR ^¥^	
Three-quarter pace		1	0–1	1	0–1	1	0–1	1	0–1	0.24
Track gallop		0	0–2	0	0–2	0	0–2	0	0–3	0.30
Jump-outs		0	0–1	0	0–1	0	0–1	0	(0–3)	0.03
Trials		0	(0–4)	0	(0–4)	0	(0–2)	0	(0–2)	0.01
Races		0	(0–7)	0	(0–3)	0	(0–7)	0	(0–2)	0.13
Total gallop days		0	0–4	0	0–5	0	0–4	0	0–3	0.12
Average days per week										
		Median	IQR ^¥^	Median	IQR ^¥^	Median	IQR ^¥^	Median	IQR ^¥^	
Three-quarter pace		1	0–1	1	0–1	1	0–1	1	0–1	0.23
Track gallop		0	0–0.4	0	0–0.3	0	0–0.4	0	0–0.5	0.47
Jump-outs		0	0–0.1	0	0–0.2	0	0–0.1	0	(0–0.5)	0.06
Trials		0	(0–0.5)	0	(0–0.5)	0	(0–0.3)	0	(0–0.3)	0.01
Races		0	(0–0.5)	0	(0–0.3)	0	(0–0.5)	0	(0–0.2)	0.13
Total gallop days		0	0–0.6	0	0–0.5	0	0–0.8	0	0–0.7	0.30
Total cumulative distance (f)										
		Median	IQR ^¥^	Median	IQR ^¥^	Median	IQR ^¥^	Median	IQR ^¥^	
Three-quarter pace		4	0–14	6	0–16	6	0–17	3	0–9	0.12
Track gallop		0	0–4	0	0–4	0	0–4	0	0–4	0.34
Jump-outs		0	0–2	0	0–4	0	0–2	0	(0–9)	0.34
Trials		0	(0–19)	0	(0–19)	0	(0–9)	0	(0–8)	0.02
Races		0	(0–49)	0	(0–15)	0	(0–49)	0	(0–12)	0.14
Total gallop distance		0	0–7	0	0–9	0	0–9	0	0–4	0.11
Average distance per week in training (f)										
		Median	IQR ^¥^	Median	IQR ^¥^	Median	IQR ^¥^	Median	IQR ^¥^	
Three-quarter pace		1	0–2.4	1.2	0–2.3	1.4	0–2.8	1	0–2	0.09
Track gallop		0	0–0.6	0	0–0.5	0	0–0.8	0	0–0.7	0.52
Jump-out		0	0–0.3	0	0–0.5	0	0–0.3	0	(0–1)	0.03
Trials		0	(0–3)	0	(0–3)	0	(0–1)	0	(0–1)	0.01
Races		0	(0–3)	0	(0–2)	0	(0–3)	0	(0–1)	0.13
Total gallop distance		0	0–1.2	0	0–1	0	0–1.6	0	0–1	0.20
Training and racing milestones										
Reached milestones										
		N	%	N	%	N	%	N	%	
Three-quarter pace										0.68
	No	147	39	41	43	58	38	48	37	
	Yes	230	61	55	57	94	62	81	63	
Track gallop										0.40
	No	229	60	58	60	87	57	84	65	
	Yes	148	40	38	40	65	43	45	35	
Jump-out										0.10
	No	266	71	63	66	103	68	100	78	
	Yes	111	29	33	34	49	32	29	22	
Trial										0.02
	No	331	88	86	90	125	82	120	93	
	Yes	46	12	10	10	27	18	9	7	
Race										0.23
	No	347	92	87	91	137	90	123	95	
	Yes	30	8	9	9	15	10	6	5	
Time to reach milestones (wks)										
		Median	IQR	Median	IQR	Median	IQR	Median	IQR	
Three-quarter pace		2	1–4	4	1–4	2	1–4	2	2–3	0.001
Track gallop		4	3–5	5	2–8	4	3–5	3	3–4	0.06
Jump-out		5	4–7	6	4–8	5	3–7	4	4–5	0.33
Trial		8	6–11	9	6–11	8	5–10	8	7–12	0.58
Race		11	8–14	13	7–15	11	8–14	9	9–10	0.37

Small trainers = trainers with a mean of <5 horses in training per week, Medium trainers = mean of 6–15 horses per week, Large = >15 horses per week, ‡ *p*-value for statistical comparison between trainer categories with significance set at α = 0.05, † = not applicable to first training preparation, Wks = weeks, f = furlongs, Slow = >15 s/furlong (<13 m/s; <800 m/min; <48 km/h), Walk = horses were only exercised on the walking machine or walked by hand, Three-quarter = 15 s/furlong (13 m/s; 800 m/min; 48 km/h), Track gallop = <13 s/furlong, (>15 m/s; >900 m/min; >55 km/h), Jump-outs = non-official trials, Trials = Official trial, Total days/distance at gallop = track gallops, jump-outs, trials and races combined, ^¥^ if IQR = 0–0 range reported (in parentheses).

**Table 2 animals-11-00928-t002:** Summary training and racing statistics for two-year-old horses in their second training preparation in Queensland, Australia.

		All Two-Year-Old Horses		Small Stables		Medium Stables		Large Stables		*p*-Value ‡
		(n = 350)		(n = 72)		(n = 119)		(n = 159)		
General training characteristics										
		Median	IQR	Median	IQR	Median	IQR	Median	IQR	
Rest period before (wks)		6	4–8	5	4–9	7	5–8	5	4–7	0.05
Pre-training (wks)		2	0–3	2	0–2	2	1–3	2	0–3	0.001
Prep length (wks)		6	4–10	8	6–12	7	4–12	5	3–8	<0.001
Low speed exercise										
Total cumulative days										
		Median	IQR ^¥^	Median	IQR ^¥^	Median	IQR ^¥^	Median	IQR ^¥^	
Slow		25	14–42	37	28–55	29	17–51	18	9–27	<0.001
Average days per week in training										
		Median	IQR ^¥^	Median	IQR ^¥^	Median	IQR ^¥^	Median	IQR ^¥^	
Slow		4	4–5	5	4–5	5	4–5	4	3–4	<0.001
Non-ridden exercise										
Total cumulative days										
		Median	IQR ^¥^	Median	IQR ^¥^	Median	IQR ^¥^	Median	IQR ^¥^	
Non-ridden		0	0–3	0	0–1	0	0–4	0	0–3	0.04
Average days per week in training										
		Median	IQR ^¥^	Median	IQR ^¥^	Median	IQR ^¥^	Median	IQR ^¥^	
Non-ridden		0	0–0.5	0	0–0.1	0	0–0.7	0	0–0.5	0.03
High-speed exercise										
Total cumulative days										
		Median	IQR ^¥^	Median	IQR ^¥^	Median	IQR ^¥^	Median	IQR ^¥^	
Three-quarter pace		6	2–11	8	5–14	6	3–11	5	2–9	0.003
Track gallop		3	0–6	3	0–7	3	1–7	3	1–6	0.37
Jump-outs		1	0–1	0	0–2	1	0–2	0	0–1	<0.001
Trials		0	0–1	0	0–1	0	0–1	0	0–1	0.87
Races		0	(0–5)	0	(0–5)	0	0–1	0	(0–3)	0.002
Total gallop days		4	1–9	4	0–9	5	2–11	4	1–8	0.06
Average days per week in training										
		Median	IQR ^¥^	Median	IQR ^¥^	Median	IQR ^¥^	Median	IQR ^¥^	
Three-quarter pace		1	1–1	1	1–1	1	0–1	1	1–1	0.36
Track gallop		0.5	0–0.8	0.3	0–0.6	0.4	0.1–0.7	0.6	0.2–1	<0.001
Jump-outs		0.1	0–0.2	0	0–0.1	0.1	0–0.2	0	0–0.2	<0.001
Trials		0	0–0.1	0	0–0.04	0	0–0.1	0	0–0.1	0.48
Races		0	(0–1)	0	(0–0.4)	0	0–0.1	0	(0–0.4)	0.01
Total gallop days		0.7	0.2–1.3	0.5	0–0.8	0.8	0.3–1.1	0.8	0.3–1.1	<0.001
Total cumulative distance (f)										
		Median	IQR ^¥^	Median	IQR ^¥^	Median	IQR ^¥^	Median	IQR ^¥^	
Three-quarter pace		14	4–26	20	12–33	14	6–30	10	3–17	<0.001
Track gallop		4	0–11	4	0–10	5	1–12	4	1–10	0.26
Jump-outs		2	0–4	0	0–4	2	0–4	0	0–2	<0.001
Trials		0	0–4	0	0–3	0	0–3	0	0–4	0.84
Races		0	(0–29)	0	(0–27)	0	0–6	0	(0–17)	0.002
Total gallop distance		8	1–20	6	0–20	10	2–27	8	1–16	0.04
Average distance per week in training (f)										
		Median	IQR ^¥^	Median	IQR ^¥^	Median	IQR ^¥^	Median	IQR ^¥^	
Three-quarter pace		2.1	1–3.1	2.3	1.2–3.5	2.3	1.2–3.3	1.9	0.8–2.9	0.04
Track gallop		0.8	0–1.4	0.4	0–0.8	0.8	0.3–1.3	1	0.3–1.6	<0.001
Jump-out		0.1	0–0.5	0	0–0.4	0.3	0–0.6	0	0–0.4	<0.001
Trials		0	0–0.3	0	0–0.2	0	0–0.3	0	0–0.4	0.47
Races		0	(0–6)	0	(0–2)	0	0–0.4	0	(0–2)	0.005
Total gallop distance		1.4	0.3–2.4	0.8	0–1.9	1.5	0.5–2.5	1.5	0.3–2.5	0.01
Training and racing milestones										
Reached milestones										
		N	%	N	%	N	%	N	%	
Three-quarter pace										0.91
	No	44	13	10	14	14	12	20	13	
	Yes	306	87	62	86	105	88	139	87	
Track gallop										0.20
	No	88	25	24	33	27	23	37	23	
	Yes	262	75	48	67	92	77	122	77	
Jump-out										<0.001
	No	172	49	37	51	37	31	98	62	
	Yes	178	51	35	49	82	69	61	38	
Trial										0.89
	No	250	71	53	74	85	71	112	70	
	Yes	100	29	19	26	34	29	47	30	
Race										0.01
	No	273	78	56	78	82	69	135	85	
	Yes	77	22	16	22	37	31	24	15	
Time to reach milestones (wks)										
		Median	IQR	Median	IQR	Median	IQR	Median	IQR	
Three-quarter pace		2	1–4	4	2–5	3	1–4	1	1–2	<0.001
Track gallop		4	2–5	6	4–7	5	3–5	3	1–4	<0.001
Jump-out		6	4–7	7	5–9	6	4–7	4	3–6	<0.001
Trial		8	6–9	9	8–13	8	7–10	6	5–8	<0.001
Race		11	8–12	11	9–14	11	9–14	9	8–10	0.07

Small trainers = trainers with a mean of <5 horses in training per week, Medium trainers = mean of 6–15 horses per week, Large = >15 horses per week, ‡ *p*-value for statistical comparison between trainer categories with significance set at α = 0.05, Wks = weeks, f = furlongs, Slow = >15 s/furlong (<13 m/s; <800 m/min; <48 km/h), Walk = horses were only exercised on the walking machine or walked by hand, Three-quarter = 15 s/furlong (13 m/s; 800 m/min; 48 km/h), Track gallop = <13 s/furlong, (>15 m/s; >900 m/min; >55 km/h), Jump-outs = non-official trials, Trials = Official trial, Total days/distance at gallop = track gallops, jump-outs, trials and races combined, ^¥^ if IQR = 0–0 range reported (in parentheses).

**Table 3 animals-11-00928-t003:** Summary training and racing statistics for two-year-old horses in their third and higher training preparation in Queensland, Australia.

		All Two-Year-Old Horses		Small Stables		Medium Stables		Large Stables		*p*-Value ‡
		(n = 349)		(n = 28)		(n = 84)		(n = 237)		
General training characteristics										
		Median	IQR	Median	IQR	Median	IQR	Median	IQR	
Rest period before (wks)		6	4–9	5	2–9	6	3–8	6	4–9	0.24
Pre-training (wks)		0	0–2	1	0–4	2	0–3	0	0–2	<0.001
Prep length (wks)		8	5–12	7	4–11	9	5–13	7	4–12	0.12
Low speed exercise										
Total cumulative days										
		Median	IQR ^¥^	Median	IQR ^¥^	Median	IQR ^¥^	Median	IQR ^¥^	
Slow		25	14–43	32	18–47	37	20–52	21	12–36	<0.001
Average days per week in training										
		Median	IQR ^¥^	Median	IQR ^¥^	Median	IQR ^¥^	Median	IQR ^¥^	
Slow		4	3–4	5	4–5	4	4–4	3	2–4	<0.001
Non-ridden exercise										
Total cumulative days										
		Median	IQR ^¥^	Median	IQR ^¥^	Median	IQR ^¥^	Median	IQR ^¥^	
Non-ridden		2	0–17	0	(0–30)	2	0–16	3	0–20	<0.001
Average days per week in training										
		Median	IQR ^¥^	Median	IQR ^¥^	Median	IQR ^¥^	Median	IQR ^¥^	
Non-ridden		0.3	0–2	0	(0–3)	0.1	0–2	0.4	0–2	<0.001
High-speed exercise										
Total cumulative days										
		Median	IQR ^¥^	Median	IQR ^¥^	Median	IQR ^¥^	Median	IQR ^¥^	
Three-quarter pace		7	3–14	7	4–13	11	5–18	7	3–12	0.003
Track gallop		4	1–8	2	0–5	6	2–11	4	2–8	0.01
Jump-outs		0	0–1	0	0–1	1	0–2	0	0–1	0.002
Trials		0	0–1	0	0–1	0	0–1	0	0–1	0.23
Races		0	0–1	0	(0–5)	0	0–2	0	(0–5)	<0.001
Total gallop days		6	2–11	2	0–8	9	2–15	6	2–10	0.003
Average days per week in training										
		Median	IQR ^¥^	Median	IQR ^¥^	Median	IQR ^¥^	Median	IQR ^¥^	
Three-quarter pace		1	1–1	1	0–1	1	1–2	1	1–1	0.01
Track gallop		0.5	0.2–0.8	0.2	0–0.5	0.6	0.3–0.8	0.5	0.3–0.8	0.001
Jump-outs		00	0–0.1	0	0–0.1	0.1	0–0.2	0	0–0.1	0.04
Trials		0	0–0.1	0	0–0.03	0	0–0.1	0	0–0.1	0.19
Races		0	0–0.1	0	(0–0.3)	0	0–0.1	0	(0–0.3)	0.001
Total gallop days		0.8	0.3–1.1	0.3	0–0.8	0.9	0.4–1.1	0.8	0.3–1.1	0.001
Total cumulative distance (f)										
		Median	IQR ^¥^	Median	IQR ^¥^	Median	IQR ^¥^	Median	IQR ^¥^	
Three-quarter pace		18	7–34	18	9–28	31	16–52	14	6–31	<0.001
Track gallop		8	2–16	2	0–9	11	3–18	8	2–16	<0.001
Jump-outs		0	0–4	0	0–3	2	0–6	0	0–3	0.002
Trials		0	0–5	0	0–2	0	0–4	0	0–5	0.15
Races		0	0–5	0	(0–30)	0	0–11	0	(0–30)	<0.001
Total gallop distance		13	3–29	4	0–18	22	4–34	12	3–28	0.01
Average distance per week in training (f)										
		Median	IQR ^¥^	Median	IQR ^¥^	Median	IQR ^¥^	Median	IQR ^¥^	
Three-quarter pace		2.3	1.3–2.6	2.5	1.3–3.8	3.6	2.2–4.4	2	1–3.1	<0.001
Track gallop		1.1	0.3–1.5	0.2	0–0.9	1.1	0.5–1.4	1.2	0.4–1.6	<0.001
Jump-out		0	0–0.4	0	0–0.4	0.2	0–0.6	0	0–0.4	0.02
Trials		0	0–0.4	0	0–0.1	0	0–0.4	0	0–0.5	0.11
Races		0	0–0.4	0	(0–2)	0	0–0.7	0	(0–2)	0.001
Total gallop distance		1.8	0.5–2.7	0.5	0–1.5	2.1	0.7–2.8	1.8	0.5–2.7	0.001
Training and racing milestones										
Reached milestones										
		N	%	N	%	N	%	N	%	
Three-quarter pace										0.54
	No	44	13	5	18	12	14	27	11	
	Yes	305	87	23	82	72	86	210	89	
Track gallop										0.06
	No	76	22	11	39	18	21	47	20	
	Yes	273	78	17	61	66	79	190	80	
Jump-out										0.02
	No	192	55	16	57	35	42	141	59	
	Yes	157	45	12	43	49	58	96	41	
Trial										0.26
	No	213	61	21	75	52	62	140	59	
	Yes	136	39	7	25	32	38	97	41	
Race										0.001
	No	256	73	24	86	49	58	183	77	
	Yes	93	27	4	14	35	42	54	23	
Time to reach milestones (wks)										
		Median	IQR	Median	IQR	Median	IQR	Median	IQR	
Three-quarter pace		2	2–4	3	2–4	2	1–3	2	2–4	0.14
Track gallop		4	3–6	5	4–7	4	3–6	4	3–5	0.06
Jump-out		6	4–7	5	4–7	6	4–7	6	4–7	0.84
Trial		8	6–10	7	5–13	9	8–10	8	6–10	0.36
Race		11	8–12	11	7–15	10	8–13	11	8–12	0.98

Small trainers = trainers with a mean of <5 horses in training per week, Medium trainers = mean of 6–15 horses per week, Large = >15 horses per week, ‡ *p*-value for statistical comparison between trainer categories with significance set at α = 0.05, Wks = weeks, f = furlongs, Slow = >15 s/furlong (<13 m/s; <800 m/min; <48 km/h), Walk = horses were only exercised on the walking machine or walked by hand, Three-quarter = 15 s/furlong (13 m/s; 800 m/min; 48 km/h), Track gallop = <13 s/furlong, (>15 m/s; >900 m/min; >55 km/h), Jump-outs= non-official trials, Trials = Official trial, Total days/distance at gallop = track gallops, jump-outs, trials and races combined, ^¥^ if IQR = 0–0 range reported (in parentheses).

## Data Availability

The data presented in this study are available on request from the corresponding author. The data are not publicly available due to privacy.

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
