# Peer review of "A Prospective Study of Training Methods for Two-Year-Old Thoroughbred Racehorses in Queensland, Australia, and Analysis of the Differences in Training Methods between Trainers of Varying Stable Sizes"

_animals, 2021, doi:10.3390/ani11040928_

Round 1

Reviewer 1 Report

REVIEW- A prospective study of training methods for Two-Year-Old Thoroughbred Racehorses in Queensland, Australia

Overall comments

General point- it would be helpful for reviewers to have line numbers for future submissions.

This is on the whole a well written article describing training practices for 2-year-old horses from data prospectively collected which has not been previously conducted in Australia and therefore provides valuable and interesting information to the racing community. There is no sample size calculation provided and I would suggest that if this was not conducted that a post-hoc calculation be provided in the discussion. The discussion also needs to include within the limitations the fact that the collected data is approximated and trainer reported, and therefore does not represent validated workloads. The continual use of “differences between trainers” needs to be replaced by something to the effect of “differences by size of stable” as there is no analysis of variance to determine differences between individual trainers. I’m concerned that the analysis is overly simplistic, and at a minimum I would suggest that the methods need to be reworded to make it explicitly clear that only univariable analysis was undertaken. However, the generation of some multivariable models adjusting for trainer/stable size, horse sex, track, preparation number etc would greatly improve on the quality of data presented here. The results are difficult to read and would benefit from the reorganisation of findings by moving some information into tables (the main manuscript currently has not presented any detail in tables) with the main messages in the text- it is very difficult to find pertinent information that is then discussed in the discussion

The figure in my version seems a bit blurry for some of the words- perhaps the editor is better suited to determine if the image is of sufficient quality to be included or whether a higher resolution file should be provided.

General note regarding the reference list- journal names should be all capitalised (currently written as e.g. “New Zealand veterinary journal”,” Equine veterinary journal” ), and theses require more detail- university, university location, date of completion .

Specific comments:

Title-  does two-year-old require capitalisation?

The title should include something to the effect of “differences in training methods for varying stable sizes” as all the results are focussed not of differences between individual trainers, but on differences between different stable sizes.

Simple summary-

Need to be clear that the training methodologies investigated for 2year-old horses only as it reads in isolation like all age groups in active race training have been investigated.

Whilst I understand that you don’t want to repeat the abstract, the purpose of the simple summary is to describe the findings of the paper in plain language. The vast majority of the summary is background which should only be a single sentence or two. There is a sentence highlighting the potential variation across different racing jurisdictions, so the line “we address this lack of data” could imply that the interviews were conducted across different locations, when in fact even within a large state these trainers were closely clustered.

There are no results presented in this summary which needs to be amended.

Abstract-

Need to include “(MSI)” in 1st sentence as used for remainder of paragraph.

Again background could be condensed.

 “how these vary between trainers, which the study sought to address.” / “Differences between trainers were evaluated using…”/” Significant differences were observed between trainers “ – not clear that you are looking at differences between different training stable sizes, not identifying differences between individual trainers.

“1) Small stables with five or less,” – should this not be 3-5 based on previous stable exclusion criteria which excluded trainers with <3 horses?

Please include the accompanying p-values in the abstract for the significant findings.

Not clear what you are referring to by “typical program”- please reword.

Introduction-

References [1-3] in isolation are inappropriate for the first sentence with the wording “remain a significant problem” as they are all >10 years old. Please include some more recent studies to substantiate this point.

2nd sentence “repetitive” used twice and again in next sentence- consider replacing 2nd (e.g. Cumulative loading)

Whilst the inclusion of the explanation of injuries occurring due to repetitive musculoskeletal loading, the introduction is quite long and some of the detail regarding collagen matrices, crimp angles etc are unnecessary to the aims of this paper, and there is no further mention of these principles in the discussion. Microdamage is a presentation of bone fatigue, but the manuscript reads as if they were two separate things and would therefore benefit from rewording. Please explain what you mean by “compete bone failure” (i.e. how is this distinct from catastrophic fracture). I would also advise rewording so that it is clear to the reader that a catastrophic fracture may occur through the coalescence of microcracks without a preceding overt stress fracture as it could otherwise be construed by an uninformed reader that Thoroughbred racehorses are being trained/raced with fractures already present, or that the signs of an imminent fracture are always present.

“Fragmented in outcome measures” – not clear, please reword.

Regarding reference 41, I would advise to remove “voluntary” as that study looked at describing voluntary rest periods in the absence of injury or other performance limiting condition- I agree that it did not account for the “involuntary” rest periods due to the differences in study design.

“Bolwell and col-leagues… unclear whether those findings are applicable to other geographic locations”- please specify the country these findings are from.

“on what constitutes an actual training program” – unclear, please reword or explain. Are you meaning what constitutes a training preparation?

Aim- need to specify that you are aiming to determine a difference in training practices between different stable sizes.

Hypothesis- difference between trainers as above- you have not tested for variance within groups. For stable size, what was the predicted direction of effect?

Methods

Why was 13 months chosen as the timeframe to monitor horses? Was there a sample size calculation for determining differences between groups? Was there any determination for the number of individual horses required? Please specify.

Per the comment regarding the abstract, please revise the definition of small trainers (or explain the difference from the companion paper).

Detailed daily exercise- slow days paragraph add space between 15 and spf.

It is unclear the numbering organisation here of 1-5- are these categories of slow days? (that was how I originally read it, but then I’m confused as to why high speed exercise in point 5 is included, galloping on treadmills would then also not be part of this section) Should there be some other prefacing sentence?

Do all the trainers use the terminology “three-quarter-pace for 15spf? In other Australian states to my knowledge ¾ pace is typically 17 spf, with 15spf = evens/ even-time/even time gallop, therefore I am surprised that all trainers would be using that terminology. Were the workouts described in the interviews in seconds or by term? This may create confusion with comparison to other studies.

Trials is an Australian racing term, it would therefore be beneficial for readers to include a description and/or reference for what this means.

Total exercise- 2.3.4/5 total distance needs a unit of measurement. If you are using furlongs with 200 m track markers please specify for international readers that they are metric equivalents of furlongs (I.e. not actually 201.168 m)

2.3.6- “active race training”- it is unclear if all trainers necessarily use pretrainers or if there are some smaller trainers doing slow work introduction from paddock fitness themselves at the training stable, and if it is the latter is this scenario included in the “active race training” phase?

 Data analysis:

Descriptive data says IQR, but some are ranges in the results- please make it clear in the methods that the IQRs of (0,0)  (?? I think) are presented as ranges instead.

I’m concerned that the analysis is overly simplistic, and at a minimum I would suggest that the methods need to be reworded to make it explicitly clear that only univariable analysis was undertaken. Did you account for clustering at the trainer level for the linear regression models? Did you consider accounting for horse sex in your analyses? Or race track? If this was done in a multivariable model then preparation number could also be accounted for.

You’ve said that where there were model specification errors you used a difference of medians instead, but what which results used this? What proportion of your results where with the different types of test for continuous outcomes.

Appendix Fig A1: Please replace current asterixis points to bullet points. Consider just using one speed term (e.g. 15spf) with an Asterix to a footnote with the alternative terms are it is quite busy.

Appendix tables- need to specify statistical test for p-values.

Results

Weekly data collection for 56 weeks is very impressive and shows a great deal of commitment and perseverance!  Were there no dropouts of trainers besides the retiree? Very obliging trainers if that is the case!!

“The overall summary training and racing statistics”- overall and summary seem superfluous, and racing statistics reads like horse form. Consider rewording to something like “Training and racing data for the 535 horses summarised by trainer are presented in Appendix Table A1. Data stratified by preparation number (whereby 377, 350 and 349 horses contributed data for preparations 1, 2 and 3 onwards, respectively) are presented in Appendixes A2-A4.”

It would also be helpful to label appendices A1-A5 rather than figure A1 and table A1.

Distances- please include spaces between number and furlongs, same for days.

Results sections 3.1 to 3.5 are extremely longwinded to follow and difficult to follow due to the large amount descriptive data. The supplementary tables are detailed but there is no tables forming the main text. These descriptive data in these sections need to be moved to a new Table 1 with summary columns as median (IQR) for small/med/large stables, so that the reader can follow the findings outlined in the main text.

Some findings as described earlier have a range rather than IQR and some say IQR (0,0) range (X,X) which is inconsistent and not clearly defined in the methods. I would suggest in a table having a footnote marker to clearly demarcate which are ranges rather than IQRs

Section 3.2.2 Please reiterate the difference between “cumulative number of furlongs exercised at a gallop” and the further down “total furlongs at a gallop” being that total = training + jumpout/trial/race, as it is confusing to read without rechecking all the methods.

Section 3.1 and 3.4 start with discussion-esque sentences saying there was significant differences. As you go on to specify what these are the first sentences can be deleted.

Section 3.5 occurs twice, please amend.

Figure 1. I like this figure, but if possible please add error bars to give some indication of variance in the population rather than a simple stacked bar chart.  Not sure that it is high enough resolution?

Discussion

Some paragraphs would benefit from restructuring- there are several short paragraphs on each subject that could be more seamlessly combined .

Pick one speed terminology and stick with it once defined- tedious to read 3-4 terms each time.

Paragraph starting with “the time taken to…” -unclear on your line “This finding emphasizes the difference be-tween the intended (“typical”) training programs reported by trainers, and the actual training programs that occur after voluntary and involuntary interruptions have occurred” The intended workload paper ([41]) you say has an equivalent time to your study, so does that not suggest that the intended and actual programs are similar? Please reword to explain.

The biggest limitation I can see is the fact that the data collected is not speed specific and the workouts aren’t stopwatched or utilising any wearable technology (e.g. GPS/accelerometers), so there is no validation of the reports of the completed workouts by the trainer. This study is obviously a considerable undertaking and therefore a useful addition to the literature, but I do think that the lack of validated workload data is something that should be explicitly acknowledged in the limitations, and it something that can be tackled in future studies.

Also a minor point with the limitations- the weekly interviews reduces recall bias, but does not completely eliminate since it is not daily data collection- please replace “Avoids recall bias” with “reduces” or “minimises” . Also regarding stable size, it would be helpful for the reader to understand here why you wouldn’t be interested in the training practices of horses with only 1-2 horses. Are these just very unlikely to have a 2yo as their 1 horse??

“This is likely to be because no fitness was lost during the short rest period.” – would benefit on the addition of some information/references for how quickly a horse is expected to lose fitness. Just because a horse has not lost CV fitness doesn’t been there aren’t consequences on the state of bones, so any references you had to support your statement would be beneficial.

Reviewer 2 Report

The study “ A prospective study of training methods for Two-Year-Old Thoroughbred Racehorses in Queensland, Australia” is really important. The approach of the study appears very original. The contents of the manuscript are quite interesting by his methodology and through the tools of quantification used. I find it interesting. I thus find that this paper definitively delivers results that will surely be of interest to the readership of the journal Animals.

However, some parts needs clarification. Because there are not Lines numbers I wrote the general comments without detailed ones.

General comments

Introduction

First of all I recommend to add some information about more common musculoskeletal injuries in race horses. In my opinion this information is missing.

In addition, there is nothing add about training monitoring in race horses. In my opinion it is extremely important to add some information about this, because trainers use them to avoid fatal injuries in the horses. The major parameter which is used for exercise monitoring is lactate blood concertation. Also other more developed techniques are nowadays used in race training such as infrared thermography which also correlates with lactate concentration. Also other parameters may be used in training monitoring such as  PBMCs proliferation and activity or cytokines mRNA expression. However, not always are they evaluated for different training methods. Thus those information should be added.

Results

The tables included in Appendix are more reader friendly than just text. I recommend to add this table as regular ones.

Discussion

It will be interesting to add information in which training regiments the musculoskeletal injuries occur the most common and why.

References

Authors should cite more recent publications because some of them are realy old (ex. 1980s).

Author Response

The study “ A prospective study of training methods for Two-Year-Old Thoroughbred Racehorses in Queensland, Australia” is really important. The approach of the study appears very original. The contents of the manuscript are quite interesting by his methodology and through the tools of quantification used. I find it interesting. I thus find that this paper definitively delivers results that will surely be of interest to the readership of the journal Animals.

However, some parts needs clarification. Because there are not Lines numbers I wrote the general comments without detailed ones.

We thank Reviewer 2 for their kind comments and have responded to their concerns below.

General comments

Introduction

First of all I recommend to add some information about more common musculoskeletal injuries in race horses. In my opinion this information is missing.

We acknowledge this deficiency and we have incorporated this information into the introduction.

In addition, there is nothing add about training monitoring in race horses. In my opinion it is extremely important to add some information about this, because trainers use them to avoid fatal injuries in the horses. The major parameter which is used for exercise monitoring is lactate blood concertation. Also other more developed techniques are nowadays used in race training such as infrared thermography which also correlates with lactate concentration. Also other parameters may be used in training monitoring such as  PBMCs proliferation and activity or cytokines mRNA expression. However, not always are they evaluated for different training methods. Thus those information should be added.

While assessing methods of monitoring the training of racehorses was beyond the scope of this paper, we acknowledge that there should be discussion around these modalities. A paragraph has been added to the discussion to address this concern. We are currently analysing survival data addressing how training methods affect the time to musculoskeletal injury in this population of racehorses and aim to publish the results in the near future.

Results

The tables included in Appendix are more reader friendly than just text. I recommend to add this table as regular ones.

Appendix tables 2, 3 and 4 have been added as regular tables rather than appendices.

Discussion

It will be interesting to add information in which training regiments the musculoskeletal injuries occur the most common and why.

I agree with Reviewer 2. However, as mentioned above, this combined information is too much for a single paper and we aim to publish detailed associations between the two-year-old training methods used in this population and musculoskeletal injuries in the nar future.

References

Authors should cite more recent publications because some of them are realy old (ex. 1980s).

While we agree with Reviewer 2 in principle, the older studies were cited to illustrate that MSI have been an ongoing problem for racehorses for an extremely long time, thus more research must focus on how training methods could mitigate this risk. I am happy to remove them if Reviewer 2 is not satisfied with this explanation.

Reviewer 3 Report

2.2.2

Who conducted the personal structured interviews?  (i.e. were they trained or not)

How were the questions validated?

2.3.1

Classification of trainers.

  • Second paragraph unnecessary as it Is not a description of methods; omit

Author Response

Who conducted the personal structured interviews? (i.e. were they trained or not)

The first author (a registered specialist equine surgeon with extensive experience in the Thoroughbred racing industry at many levels) conducted the structured interviews every week.

How were the questions validated?

We validated the questions through a series of pilot studies. The terminology used in the questions and units used to report the data were clarified at the commencement of the study.

2.3.1

Classification of trainers.

  • Second paragraph unnecessary as it Is not a description of methods; omit

We respectfully request to leave this paragraph in, as although it is not strictly methods, it provides an explanation of our methodology to help the reader understand why these categories were chosen. If this explanation is not acceptable then we are happy to delete the paragraph.

Reviewer 4 Report

The manuscript describes data about two-year-old thoroughbreds training methodologies and highlights differences in training volume and preparation characteristics between trainers in Queensland, Australia. 

Information obtained could be useful for developing training and management strategies to mitigate the risk of musculoskeletal injuries in young racehorses in Australia.

In my opinion the introduction is concise and provides adequate background. The study design is appropriate, and methods are well described. Although results are clearly reported, I find that tables contain a lot of data and can be difficult to interpret. Conclusions are appropriate and supported by results.

Specific comments:

Results:

I agree with the authors who state that knowledge of what constitutes a true training program, which incorporates planned and unplanned interruptions, could be important in order to modulate two-year-old racehorses training program and reduce the risk of musculoskeletal injuries. However, detailed data on rest periods in terms of duration and motivation are not reported in the present study. I think such information might be of interest to the reader. Are rest periods due to planned or unplanned interruptions? I suggest including and discussing them.

Discussion:

There is a repetition “… New South Wales, Australia [38] but lower than lower than that reported in New Zealand [40,64].”

The authors commented that approximately 40% of the two-year-old horses in their study started in a trial and 30% started in a race and these results are very low compared to those reported in other studies. The authors hypothesized that the differences are due to the high percentage of horses from elite yearling sales included in previous studies. However, the percentage of high-quality young racehorses in the present study is not available. How can they explain such big difference? are there other possible explanations? is it possibly due to musculoskeletal injuries or other causes of unplanned interruptions of training? 

Author Response

The manuscript describes data about two-year-old thoroughbreds training methodologies and highlights differences in training volume and preparation characteristics between trainers in Queensland, Australia. 

Information obtained could be useful for developing training and management strategies to mitigate the risk of musculoskeletal injuries in young racehorses in Australia.

In my opinion the introduction is concise and provides adequate background. The study design is appropriate, and methods are well described. Although results are clearly reported, I find that tables contain a lot of data and can be difficult to interpret. Conclusions are appropriate and supported by results.

I would like to thank Reviewer 3 for their positive review and helpful comments and hopefully all of your concerns are addressed in the revised manuscript. I struggled with how to convey such a large amount of data in an easy to understand format. The descriptions in the text were aiming to highlight the most important findings. I am not sure how I can simplify these tables without losing important information.

Specific comments:

Results:

I agree with the authors who state that knowledge of what constitutes a true training program, which incorporates planned and unplanned interruptions, could be important in order to modulate two-year-old racehorses training program and reduce the risk of musculoskeletal injuries. However, detailed data on rest periods in terms of duration and motivation are not reported in the present study. I think such information might be of interest to the reader. Are rest periods due to planned or unplanned interruptions? I suggest including and discussing them.

I agree that it would have been better to present the reasons why the training preparations were ended. Unfortunately, collecting detailed information regarding the exact reason for the rest periods in this study was beyond our resources. We are currently analysing survival data on the time to musculoskeletal injury and evaluating how many training practices including rest periods affect the time to musculoskeletal injury.

Discussion:

There is a repetition “… New South Wales, Australia [38] but lower than lower than that reported in New Zealand [40,64].”

Noted and amended.

The authors commented that approximately 40% of the two-year-old horses in their study started in a trial and 30% started in a race and these results are very low compared to those reported in other studies. The authors hypothesized that the differences are due to the high percentage of horses from elite yearling sales included in previous studies. However, the percentage of high-quality young racehorses in the present study is not available. How can they explain such big difference? are there other possible explanations? is it possibly due to musculoskeletal injuries or other causes of unplanned interruptions of training?

Understood. The discussion addressing these concerns has been expanded.

Round 2

Reviewer 1 Report

The manuscript on the whole has addressed my initial concerns.

My remaining concern is still in regards to the results layout. Based on other reviewers suggestions the tables in the Appendix have now been included in the main text. However the 4 tables are very large and have been placed all at the end of the results section when the journal convention would be for the tables to include the tables shortly after they are referred to in the text. The results text has not been condensed and therefore there is substantial duplication of the table material in the results text, which should be addressed prior to publication. 

Also a minor point that might be better assessed by the editorial team, but journal names are still not all capitalised. I would advise if it is endnote that is causing the disturbance then to just turn endnote off and manually alter. 

Simple summary wording "We investigated training method in Queensland, Australia"- should this be "methods"? In any case this sentence is superfluous to next sentence of "we investigated the variation in training methodologies" 

Author Response

The manuscript on the whole has addressed my initial concerns.

Thank you, we appreciate your feedback.

My remaining concern is still in regards to the results layout. Based on other reviewers suggestions the tables in the Appendix have now been included in the main text. However the 4 tables are very large and have been placed all at the end of the results section when the journal convention would be for the tables to include the tables shortly after they are referred to in the text. The results text has not been condensed and therefore there is substantial duplication of the table material in the results text, which should be addressed prior to publication. 

The results have been condensed to reduce the duplication.

Also a minor point that might be better assessed by the editorial team, but journal names are still not all capitalised. I would advise if it is endnote that is causing the disturbance then to just turn endnote off and manually alter. 

Noted and amended.

Simple summary wording "We investigated training method in Queensland, Australia"- should this be "methods"? In any case this sentence is superfluous to next sentence of "we investigated the variation in training methodologies" 

Understood- removed.